# Endocannabinoid System Components of the Female Mouse Reproductive Tract Are Modulated during Reproductive Aging

**DOI:** 10.3390/ijms24087542

**Published:** 2023-04-19

**Authors:** Gianna Rossi, Valentina Di Nisio, Alessandro Chiominto, Sandra Cecconi, Mauro Maccarrone

**Affiliations:** 1Department of Life, Health and Environmental Sciences, University of L’Aquila, 67100 L’Aquila, Italy; gianna.rossi@univaq.it; 2Department of Gynecology and Reproductive Medicine, Karolinska University Hospital, SE-14186 Stockholm, Sweden; valentina.di.nisio@ki.se; 3Division of Obstetrics and Gynecology, Department of Clinical Science, Intervention and Technology, Karolinska Institutet, SE-14186 Stockholm, Sweden; 4Department of Pathology, San Salvatore Hospital, 67100 L’Aquila, Italy; alessandro.chiominto@univaq.it; 5Department of Biotechnological and Applied Clinical Sciences, University of L’Aquila, 67100 L’Aquila, Italy; 6European Center of Brain Research, Santa Lucia Foundation IRCCS, 00164 Rome, Italy

**Keywords:** endocannabinoid system, ovary, oviduct, uterus, aging, expression and distribution

## Abstract

The endocannabinoid (eCB) system has gained ground as a key modulator of several female fertility-related processes, under physiological/pathological conditions. Nevertheless, its modulation during reproductive aging remains unclear. This study aimed to investigate the expression levels of the main receptors (cannabinoid receptor 1,CB_1_; cannabinoid receptor 2, CB_2_; G-protein coupled receptor, GPR55; and transient receptor potential vanilloid type 1 channel, TRPV1) and metabolic enzymes (N-acylphosphatidylethanolamine phospholipase D, NAPE-PLD; fatty acid amide hydrolase, FAAH; monoacylglycerol lipase, MAGL; and diacylglycerol lipase, DAGL-α and -β) of this system in the ovaries, oviducts, and uteri of mice at prepubertal, adult, late reproductive, and post-reproductive stages through quantitative ELISA and immunohistochemistry. The ELISA showed that among the receptors, TRPV1 had the highest expression and significantly increased during aging. Among the enzymes, NAPE-PLD, FAAH, and DAGL-β were the most expressed in these organs at all ages, and increased age-dependently. Immunohistochemistry revealed that, regardless of age, NAPE-PLD and FAAH were mainly found in the epithelial cells facing the lumen of the oviduct and uteri. Moreover, in ovaries, NAPE-PLD was predominant in the granulosa cells, while FAAH was sparse in the stromal compartment. Of note, the age-dependent increase in TRPV1 and DAGL-β could be indicative of increased inflammation, while that of NAPE-PLD and FAAH could suggest the need to tightly control the levels of the eCB anandamide at late reproductive age. These findings offer new insights into the role of the eCB system in female reproduction, with potential for therapeutic exploitation.

## 1. Introduction

Female fertility problems can be determined by several causes, such as ovulation disorders, uterine abnormalities, fallopian tube alteration, and metabolic diseases. In recent years, aging has also been considered an important cause of infertility because many women are delaying pregnancy, and this postponement involves a decline in the quantity and quality of ovulated oocytes and embryos, together with a higher risk of obstetrical complications [1]. Thus, a successful pregnancy requires many quality checks that are technically more challenging to assess than oocyte number.

Among the many paracrine and endocrine factors controlling fertility, gonadotropins (FSH and LH) together with steroids (estrogens and progesterone) are considered the main regulators of the ovarian cycle, pregnancy, and menopause [2]. In recent years, there has been increasing interest in endocannabinoids (eCBs) as critical modulators of the female reproductive processes following the discovery of metabolic enzymes and receptors of the eCB system (ECS) in ovaries, oviducts, and uteri [3,4]. In line with this, accumulated evidence demonstrates the pivotal role of ECS in regulating female reproductive events such as oocyte maturation, fertilization, and both early and late pregnancy in a species-specific manner [4]. Indeed, the presence of eCBs and of their receptors and metabolic enzymes in reproductive fluids, cells, and tissues strengthen the hypothesis that tuning eCB signaling could be essential to warrant successful reproduction.

In human ovaries, El-Talatini and colleagues [5,6] demonstrated that the eCBs anandamide (AEA) and 2-arachidonoylglycerol (2-AG), along with their metabolic enzymes, fatty acid amide hydrolase (FAAH, responsible for AEA degradation) and monoacylglycerol lipase (MAGL, responsible for 2-AG degradation), are mainly expressed in ovarian somatic cells. In mice, preovulatory germinal vesicle oocytes expressed the four major receptors (namely, cannabinoid receptor 1 and 2, CB_1_ and CB_2_; G-protein coupled receptor 55, GPR55; transient receptor potential vanilloid type 1 channel, TRPV1) that underwent different meiotic maturation-dependent changes in their expression [7]. In addition, in rats [8] and cats [9], the presence of FAAH has been extensively documented in oocytes within the follicles at diverse maturation stages. Moreover, the overstimulation of CB_2_ induced massive oocyte apoptosis in fetal ovaries, thereby depleting ovarian reserve [10].

In the oviducts, AEA, CB_1_, CB_2_, FAAH, and N-acylphosphatidylethanolamine phospholipase D (NAPE-PLD), the key biosynthetic enzyme of AEA [4], have a crucial role in the fertilization process and then in the transport of the developing embryos to the uterus [11,12]. It is well known that the maintenance of a tightly controlled AEA tone and a low expression of CB_1_ receptors in the oviducts are correlated with normal embryo movement down to the uterus and reduced risk of ectopic pregnancy, respectively [13]. In addition, the fine tuning of AEA tone in oviducts is essential for regulating sperm capacitation via CB_1_ and TRPV1 [14], an observation that was confirmed by the sperm-targeted deleterious effect of high levels of AEA in *faah^−/−^* mice [15].

ECS also modulates endometrium plasticity and uterus receptivity because the synchronization between the developing embryo and endometrium is AEA-dependent. Overall, the tight control of AEA (and 2-AG) concentration is recognized as a pivotal clue to implantation, decidualization, and placentation, under the regulation of progesterone and estrogen [16,17]. Interestingly, the AEA levels are low in receptive uterus and implantation areas, but high in non-receptive uterus and in extra-implantation areas [18]. Altogether, these data indicate a crucial role of ECS in the success of the whole reproductive process.

To date, the variation in the main ECS components during female reproductive aging has not yet been interrogated. In this study we sought to fill this knowledge gap by using a mouse model to analyze the protein expression levels of the main receptors (CB_1_, CB_2_, GPR55, and TRPV1) and enzymes (NAPE-PLD, FAAH, MAGL, and diacylglycerol lipases (DAGL)-α and -β that synthesize 2-AG) of ECS, together with their tissue localization. The aim was to explore, for the first time, the differential expression and tissue localization of these key ECS elements in the female reproductive tract during physiological aging.

## 2. Results

### 2.1. ECS Component Quantification in Female Reproductive Tract during Aging

To investigate the age-dependent effects on the expression of the main ECS receptors and enzymes in the female mouse reproductive tract, the whole panel of ECS proteins was determined by means of quantitative ELISA. Ovaries, oviducts, and uteri from mice at different ages, i.e., prepubertal (PrP; 6–11 days old), adult (Ad; 3–5 months old), late reproductive (LR; 9–12 months old), and post-reproductive (PR; >15 months old), were collected and processed for assessing ECS components protein expression levels. As shown in Figure 1, the eCBs-binding receptors CB_1_, CB_2_, and GPR55 had a generally low expression in all of the organs investigated.

As shown in Figure 1a and in Appendix A, TRPV1 was significantly upregulated in the PR ovaries compared to all of the other age groups (*p* < 0.001 vs. PrP, Ad, LR; Figure 1a and Appendix A); contrastingly, the CB_1_ and CB_2_ receptors reached their highest ovarian expression in the LR stage (0.099 ± 0.01 and 0.059 ± 0.001, respectively) compared to all of the other age groups (*p* < 0.05, Figure 1a and Appendix A). These results were mirrored by the expression of NAPE-PLD, which was elevated at all ages, but increased significantly in the LR groups (0.527 ± 0.02; *p* < 0.05 vs. PR; *p* < 0.001 vs. Ad, Figure 1a and Appendix A). Concomitantly, the expression of the AEA degrading enzyme FAAH was inversely related to that of NAPE-PLD, having its highest and lowest expression in PR and LR mice, respectively (0.390 ± 0.02 vs. 0.251 ± 0.01; *p* < 0.05, Figure 1a and Appendix A). Similarly, DAGL-β was significantly upregulated in the PR ovaries compared to all of the other age groups (0.223 ± 0.01; *p* < 0.001 vs. PrP, Ad and LR, Figure 1a and Appendix A). As for GPR55, DAGL-α, and MAGL, the ovarian values were comparable and were unaffected by aging (*p* > 0.05, Figure 1a and Appendix A).

In the mice oviducts, CB_1_ displayed a significant increase at the LR stages in comparison with the other experimental groups (0.083 ± 0.001; *p* < 0.01 vs. PrP; *p* < 0.001 vs. Ad, PR, Figure 1b and Appendix A). Mirroring CB_1_, the expression of NAPE-PLD was also significantly upregulated in the LR oviducts (0.5183 ± 0.01; *p* < 0.05 vs. Ad, PR, Figure 1b and Appendix A). By contrast, the FAAH values showed comparable levels in all age groups (*p* > 0.05, Figure 1b and Appendix A). Interestingly, all of the main enzymes of 2-AG metabolism, as well as a key 2-AG-binding receptor, appeared to have a similar trend in the oviducts during reproductive aging. Indeed, the detection of DAGL-α, MAGL, and CB_2_ documented a high expression during the PrP age in comparison to all of the other age groups (*p* < 0.001 vs. Ad, LR and PR, Figure 1b and Appendix A). On the other hand, DAGL-β and GPR55 remained stable up to the LR age and only showed a sharp increase in the oviducts of the PR mice (*p* < 0.001 vs. PrP, Ad, LR, Figure 1b and Appendix A). Finally, TRPV1 also appeared to be significantly upregulated in the PR age group compared with the younger counterparts (0.18 ± 0.001; *p* < 0.001 vs. PrP, Ad, LR, Figure 1b and Appendix A).

Overall, the expression of the main ECS components in the mice uteri showed a predominant modulation of AEA-related receptors and metabolic enzymes during aging. Regarding the AEA metabolic enzymes, NAPE-PLD was significantly upregulated in the uteri of LR mice (0.641 ± 0.04, *p* < 0.05 vs. PrP, PR; *p* < 0.01 vs. Ad, Figure 1c and Appendix A), while FAAH underwent an exponential increase in the uteri collected from the Ad to PR age groups (0.248 ± 0.02 vs. 0.377 ± 0.01, respectively; *p* < 0.001, Figure 1c and Appendix A). GPR55 was significantly upregulated during the PR stage compared to younger ages (0.064 ± 0.001; *p* < 0.05 vs. PrP, Ad, LR, Figure 1c and Appendix A). Conversely, the key 2-AG-binding receptor CB_2_ appeared to be highly expressed in the PrP stage uteri (0.065 ± 0.001; *p* < 0.001, Figure 1c and Appendix A). DAGL-α and -β showed a significant downregulation in the uteri collected from the Ad mice in comparison with those from other age groups (DAGL-α: 0.116 ± 0.01; *p* < 0.01 vs. PR, *p* < 0.001 vs. PrP, LR; DAGL-β: 0.096 ± 0.01; *p* < 0.01 vs. PrP, LR, PR, Figure 1c and Appendix A). Finally, the remaining ECS components, TRPV1 and MAGL, failed to show a significant change in the uteri during aging (*p* > 0.05, Figure 1c and Appendix A).

### 2.2. Tissue Localization of ECS Components in Female Reproductive Tract during Aging

Based on the ELISA results, we chose to perform an immunohistochemistry (IHC) analysis of the AEA metabolic enzymes, NAPE-PLD and FAAH, in order to ascertain their tissue localization in mouse ovaries, oviducts, and uteri during aging.

In the mouse PrP ovaries, both NAPE-PLD and FAAH were detected predominantly in the ovarian surface epithelium (Figure 2a,b). Interestingly, NAPE-PLD was also localized in the interstice between the granulosa cells and oocytes of early-stage follicles, up to the preantral stage (Figure 2a). During the Ad stage (Figure 2e), the enzyme localization in the oocyte cytoplasm started to increase until reaching its maximum expression at the LR stage, where its immunoreactivity was most evident in the ovarian surface epithelium, oocytes of small early-stage follicles, and granulosa cells in the follicles from the secondary stage onward (Figure 2i). Conversely, FAAH distribution appeared to be very homogeneous in the ovaries of Ad mice (Figure 2f), being detected in both oocytes and granulosa cells (mainly in the foci matrix) of follicles up to the early antral stage and scattered in the surrounding stromal cells. In LR mice ovaries, the detection was more evident in the stromal cells and in the oocytes (Figure 2j). Lastly, in the ovaries of PR mice, a strong signal was present in all of the sections (Figure 2n), mirroring the overexpression detected via the ELISA assay.

The immunodetection of NAPE-PLD and FAAH in mouse oviducts showed the presence of the AEA metabolic enzymes predominantly in the outer layer of serosa and in the oviductal luminal compartment of all groups (Figure 3). As demonstrated by protein quantification, the NAPE-PLD signal appeared stronger in the PrP oviducts (Figure 3a), while the FAAH signal was more intense in the PR group (Figure 3n). It is worth mentioning that, while NAPE-PLD was found predominantly in the luminal epithelial ciliated cells of the oviducts in the PrP, Ad, and LR age groups (Figure 3a,e,i), FAAH showed a stronger signal in both the secretory and ciliated cells of the oviducts in all of the age groups (Figure 3b,f,j,n). Furthermore, the oviductal localization of NAPE-PLD in PR mice was detected mainly in the secretory cells in the luminal part, and in the outer layer of the serosa (Figure 3m).

As for the uterine expression and tissue localization of NAPE-PLD, while at the PrP age, the main AEA-synthetizing enzyme prevailed in the perimetrium (Figure 4a), during the Ad stage, its localization was focused on the glandular compartment of the endometrial glands and on the simple columnar epithelium facing the endometrial lumen (Figure 4e). Its localization became more homogeneous and dispersed in the uteri of LR and PR mice, showing signals in both glands and stromal compartments (Figure 4i,m). A similar distribution was also found by investigating the endometrial localization of FAAH. The main differences can be observed in the enzyme’s homogeneous localization in the PrP endometrium, where FAAH was also present in the mucinous compartment of the uterus, but not in the myometrium (Figure 4b). Furthermore, in uteri collected from the PR mice, a strong FAAH signal was shown in the epithelial, stromal, glandular, and myometrium regions close to the external perimetrium (Figure 4n). In both Ad and LR age groups, FAAH was mainly localized in the glandular and epithelial areas, with a faint positivity also being detected in part of the stromal compartment (Figure 4f,j).

Overall, the intensity presented by the immunodetection on the tissue sections of both NAPE-PLD and FAAH mirrored the data obtained by the quantitative ELISA assays.

## 3. Discussion

The ECS is well known for playing a crucial role in the physiological and pathological function of the central nervous system and immune system [19,20,21]. During the last decades, the accumulated evidence in the literature has demonstrated the influence of ECS in peripheral organs, as well the impact of eCBs as key modulators of the reproductive system [4]. Yet, systematic studies regarding ECS in the different organs of male and female reproductive tracts are still missing from its broad role identification. To fill this gap, the purpose of this study was to fully characterize the protein expression of the main ECS components, together with the tissue localization of the most expressed elements (NAPE-PLD and FAAH) in mice ovaries, oviducts, and uteri. In addition, we determined the age-dependent and organ-specific modulation of ECS during reproductive aging, i.e., prepubertal, adult, late reproductive, and post-reproductive experimental groups.

In mouse ovaries, the AEA-related CB_1_ receptor and metabolic enzymes (NAPE-PLD, and FAAH) oversee the main modulation detected during reproductive aging. Even though the species-specific expression and distribution of ECS among mammals has been revealed [4], the identification of CB_1_ and FAAH in mouse ovaries is consistent with the data reported in both rats and humans [5,8]. Within the ovary, the AEA-related ECS seems to be importantly upregulated during the LR stage, as evidenced by the significant increase in both CB_1_ receptors and NAPE-PLD enzymes. This increase, together with the localization of the NAPE-PLD in oocytes and granulosa cells up to secondary follicles, suggests that this functionally coupled structure is ready to activate AEA signaling in both a paracrine and autocrine manner, with the main target being the oocyte. This hypothesis is supported by the detection of AEA in follicular fluid [6,22] and of CB_1_ in GV-stage oocytes during the whole process of folliculogenesis [7], although it was drastically reduced after oocyte meiotic resumption. Together with the sharp increase in CB_1_, we also registered a small, but significant rise in CB_2_ in the LR mouse ovaries, even though the 2-AG metabolic enzymes (DAGL-α and MAGL) remained unchanged through reproductive aging. Of note, in CB_2_ knockout models, both ovaries and oocytes were able to maintain the functionality and competence that were highly impaired in CB_1_ knockout ovaries [23,24]. Thus, we hypothesize that in the ovarian context, the slight increase in CB_2_ could be justified as an attempt of the ECS to provide functional support to CB_1_ activity. Finally, it is noteworthy that the localization of FAAH in the late reproductive mouse ovaries is excluded from the follicular compartment and confined to the surrounding stroma. Such an unprecedented relocation of the main AEA-degrading enzyme suggests, together with the previous findings, a fine tuning of AEA activity in the follicular microenvironment, which could be necessary to support the production of high-quality oocytes.

In the oviducts of reproductively aging mice, the main modulation was observed in the 2-AG-related ECS components (CB_2_ receptor, and DAGL-α and MAGL enzymes). Indeed, these proteins display a significant upregulation in the oviducts of prepubertal mice, while their expression drops and then remains unchanged from adulthood to the post-reproductive stage. The downregulation of DAGL-α can be explained by the need to deplete 2-AG from the oviducts, thus avoiding its detrimental effect on sperm motility, capacitation, and acrosome reaction [25]. Interestingly, a higher level of NAPE-PLD—the main AEA biosynthetic enzyme—has been detected in both the ovaries and oviducts of PrP mice. Based on the strictly connected mechanisms of the whole female reproductive tract, we can speculate that high NAPE-PLD in PrP mice may warrant high AEA concentration; then, the overexpression of FAAH—the main AEA hydrolase—during later reproductive stages will be able to reduce the AEA concentration, thus ensuring a suitable gradient for embryo tubal transport towards the uterine site of implantation [26]. Moreover, the potential role of the hormonal regulation of NAPE-PLD, most likely at the promoter level, has not been investigated yet, leaving open the question of its possible impact on reproductive events.

A change in the ECS components also seems to occur in uteri, where a differential expression of both 2-AG- and AEA-related receptors and enzymes is evident during reproductive aging. For instance, the sharp increase in CB_1_ during adulthood correlates to its functional endometrial role during the preimplantation stage [27]. Moreover, the modulation of NAPE-PLD, FAAH, and DAGL-α is consistent with accumulated literature data that show a key role of eCBs in the regulation of several steps during the estrous/menstrual cycle [4,28], as well as during early and late stages of pregnancy [29,30] in several mammalian species. Interestingly, the tissue localization of both NAPE-PLD and FAAH in the oviducts and uteri appears mainly in the epithelial cells facing the lumen of the organ, or glandular cells, as also reported in other species [28,29,31,32]. The taxonomical recurrent identification of synthetizing and degrading AEA enzymes in close proximity to the reproductive organs’ lumen sheds light on the importance of balancing the AEA tone in the areas where the embryo will be transported and implanted. Indeed, an unbalanced regulation of AEA tone, mainly in the uterus, could have major consequences on the initiation and successful outcome of pregnancy [29,32,33]. In this context, it is worth mentioning that cross-talks between several ECS elements (i.e., CB_1_ and FAAH) and hormones responsible for female reproduction have been described, particularly in the uterus during placentation, pregnancy, and labor [4]. For instance, a deficiency of the *cnr1* gene, that encodes for CB_1_, leads to a high risk of preterm birth linked to an altered progesterone/estrogen ratio [34]. These findings, together with an overall reduction in NAPE-PLD and increase in FAAH expression during mouse reproductive aging, prompt us to implicate AEA as the main eCB to be depleted from ovaries, oviducts, and uteri, likely to avoid the development of pathological conditions such as cancer [35].

Another intriguing result of the present study is the age-dependent overexpression of TRPV1 receptors through the reproductive system of PR mice. The literature data on the modulation of the expression and function of the family of vanilloid receptors during aging seem to be conflicting. For instance, in the trigeminal ganglion neurons, TRPV1 expression undergoes a sharp decrease in aged mice, while TRPV2 increases in an age-dependent manner, thus modifying the sensitivity in the oral mucosa [36]. On the other hand, the upturning effect of aging on the anti-inflammatory function of TRPV1 denoted its impaired role in the suppression of tumor necrosis factor α in old mice [37], and of numerous cytokines involved in skin aging [38]. These molecular and functional modifications are found in combination in old mice testes, where a transcriptional increase in TRPV1 is associated with enhanced testicular apoptosis [39]. Against this background, a similar behavior cannot be ruled out in the female reproductive tract, and therefore a joint proinflammatory action mediated by TRPV1 and DAGL-β can be suggested to be similarly upregulated in a reproductively aged model, and to be similarly involved in the response to tumor necrosis factor α [40]. In addition, the age-related slight increase in GPR55 levels in PR mice oviducts and uteri could be connected to the role of this receptor in the regulation of insulin action and adipogenesis [41]. Since aging relates to adipogenesis and insulin sensitivity [42,43], it could be assumed that the increase in GPR55 at older ages could be instrumental to counteract age-related dysfunctions.

Altogether, the age-related modulation of ECS in the female reproductive tract points to a general reduction of AEA tone to a minimum at the PR stage, via decrease in NAPE-PLD and increase in FAAH. Instead, 2-AG metabolism seems to be organo-specific, denoting the low presence of 2-AG metabolic enzymes in the ovaries and a decrease in oviducts and uteri at the LR and PR stages. Incidentally, a FAAH increase could also contribute to the hydrolysis of 2-AG, thus justifying the lack of effect on DAGL-α expression, especially in oviducts. These potential regulations are shown in Figure 5.

## 4. Materials and Methods

### 4.1. Chemicals

The chemicals used in this study were obtained from the following companies: secondary goat anti-mouse IgG conjugated to HRP (sc-2005) from Santa Cruz Biotechnology (Santa Cruz, CA, USA), and goat anti-rabbit IgG conjugated to horseradish peroxidase (HRP), (cat. 111-035-003) from Thermo Fisher Scientific (Waltham, MA, USA). The mouse-to-mouse HRP ready-to-use kit was obtained from ScyTek Laboratories, Inc. (Logan, UT, USA). All of the other reagents and ABTS (2,20-azinobis (3-ethylbenzothiazoline-6-sulfonic acid)-diammonium salt) were purchased from Sigma (St. Louis, MO, USA) and were of the purest analytical grade. Rabbit anti-CB_1_ (cat. 101500), anti-CB_2_ (cat. 101550), anti-GPR55 (cat. 10224), anti-FAAH (cat. 101600), anti-MAGL (cat. 100035), and anti-NAPE-PLD (cat. 10305) polyclonal antibodies were obtained from Cayman Chemicals (Ann Arbor, MI, USA). Rabbit anti-DAGL-α (cat. PA5-23765) and anti-DAGL-β (PA5-26331) polyclonal antibodies were obtained from Invitrogen (Thermo Fisher Scientific; Waltham, MA, USA). The rabbit anti-TRPV1 (cat. TA336871) polyclonal antibody was obtained from OriGene Technologies (Rockville, MD, USA).

### 4.2. Animals and Sample Collection

Female Swiss CD1 mice (Charles River Laboratories, Lecco, Italy) were housed in an animal facility under controlled temperature (21 ± 1 °C) and light (12 h light/day) conditions, with ad libitum access to food and water. The ages of each animal group were chosen according to previous studies [44,45]. In brief, the animals (*n* = 28) were sorted into four groups: prepubertal (6–11 days; PrP), adult (3–5 months, Ad), late reproductive (9–12 months, LR), and post-reproductive (>15 months, PR). At the selected age, the mice were euthanatized, and the ovaries, oviducts, and uteri were either stored at −196 °C under liquid nitrogen for further analysis or fixed in 4% paraformaldehyde overnight (o.n.) at 4 °C for paraffin embedding. All of the experiments were carried out in conformity with national and international laws and policies (European Economic Community Council Directive 86/609, OJ 358, 1 Dec 12, 1987; Italian Legislative Decree 116/92, Gazzetta Ufficiale della Repubblica Italiana n. 40, Feb 18, 1992; National Institutes of Health Guide for the Care and Use of Laboratory Animals, NIH publication no. 85–23, 1985). The project was approved by the Italian Ministry of Health and the internal Committee of the University of L’Aquila in 2018–2021 (authorization number CE5C5.N.FN to SC). All efforts were made to minimize suffering. The method of euthanasia consisted of an inhalant overdose of carbon dioxide (CO_2_, 10–30%), followed by cervical dislocation.

### 4.3. ELISA Assay

The protein expression of the ECS elements was assessed using enzyme-linked immunosorbent assay (ELISA), as previously reported [46]. Briefly, wells were plated with ovaries, oviducts, or uteri lysates of the different experimental groups (5 μg/well) in coating buffer (0.05 M Na2CO3, pH = 9.6) for 2 h at room temperature. Then, they were incubated for 1 h in 1% bovine serum albumin (BSA) in phosphate-buffered saline (PBS) and then for 2 h with: anti-CB1 (1:500); anti-CB2 (1:500); anti-GPR55 (1:1250); anti-TRPV1 (1:2500); anti-NAPE-PLD (1:500); anti-FAAH (1:500); anti-DAGL-α (1:1250); anti-DAGL-β (1:2500); and anti-MAGL (1:500). All of the rabbit polyclonal antibodies were diluted in 1% BSA in PBS. The specificity of the antibodies was checked in mouse by Western blotting, as previously reported [40]. After rinsing three times with 1% BSA in PBS–Tween 20, 100 μL of horseradish peroxidase (HRP)-conjugated secondary antibody (diluted 1:5000) was added and the ELISA plate was incubated for 1 h at room temperature. Lastly, the enzymatic activity of HRP was determined by adding ABTS (100 μL), followed by 1% SDS (100 μL) to stop the reaction. A Multiskan ELISA Microplate Reader (ThermoLabsystems, Bevery, MA, USA) with absorbance at 405 nm was used for measurement. The linearity ranges of the ELISA for the used antibodies were ascertained by dose-dependence curves with different amounts (0–2.5–5.0–10 μg/well) of mouse uterus extracts (Ad) for each antibody. All data were within these linearity ranges and were expressed as absorbance units.

### 4.4. Hematoxylin–Eosin (H&E) and Immunohistochemistry (IHC)

The whole ovaries, oviducts, and uteri of mice at different reproductive ages were processed and stained for H&E assessment, as previously reported [45]. In brief, after fixation in 4% formalin, all tissues were embedded in paraffin, sectioned (4 μm/section), stained, and mounted. The sections were examined using StereoZoom^®^ Leica S8 APO and images were acquired with a Leica EC3 camera (Leica Microsystems, Wetzlar, Germany). To establish the localization of NAPE-PLD and FAAH, IHC was performed, as briefly described below. The samples were embedded in paraffin and sectioned (4 μm); after deparaffinization, the sections were rehydrated, treated with 10 mM sodium citrate (pH 6.0), and washed many times in phosphate buffered saline (PBS) for 5 min. A mouse-to-mouse HRP ready-to-use kit was used in line with the instruction manual. The sections were then incubated at 4 °C o.n. in a humidified chamber with the subsequent rabbit primary antibodies: anti-NAPE-PLD (1:120) and anti-FAAH (1:40). Negative controls were performed by incubating sections with 3% BSA. Whole tissue sections were stained simultaneously for all markers to avoid any technical bias. After washing with PBS, the slides were incubated with Ultra Tek anti-polivalent from the mouse-to-mouse kit for 15 min at r.t. After washing with PBS, the slides were incubated with Ultra Tek HRP, washed once more with PBS, and incubated with DAB for 5 min. Hematoxylin was used for counterstaining. All slides were observed using a ZeissAxio Imager A2 microscope (Carl Zeiss Microscopy Deutschland, GmbH, Oberkochen, Germany) and captured using IM500 software. Every experiment was repeated in three different biological replicates.

### 4.5. Statistical Analysis

All experiments were performed at least three times, and data were expressed as the mean ± SEM. The experimental results were analyzed using ANOVA followed by the Holm–Sidak post hoc test. The results were considered statistically significant when *p* < 0.05. All statistical analyses were performed using the statistical package SigmaPlot v.11.0 (Systat Software Inc., San Jose, CA, USA).

## 5. Conclusions

In conclusion, our investigation reveals an unprecedented modulation of ECS components in the mouse female reproductive tract during aging, showing a major effect on eCB metabolic enzymes. To the best of our knowledge, this is the first time that a systematic study has been performed on the role of reproductive aging on the ECS, avoiding the bias due to external hormonal stimulation. The localization of NAPE-PLD and FAAH in epithelial cells and lumen-surrounding cells provides another tile in the as-yet unfinished mosaic of AEA’s role in female reproductive events. Indeed, our data stress the importance of eCB signaling during reproductive ages, while strengthening the need for its downregulation in the post-reproductive stage. In this context, our results are in line with those obtained in other mammals, reinforcing the hypothesis that eCBs are broadly involved in regulating female reproduction. Furthermore, the knowledge of ECS-dependent repro-modulation could contribute to the development of drugs of therapeutic potential [47]. On a final note, it could be of interest to further investigate the role of TRPV1 and DAGL-β for dissecting the involvement of AEA-related versus 2-AG-related signaling in pathophysiology of female reproduction.

## Figures and Tables

**Figure 1 ijms-24-07542-f001:**
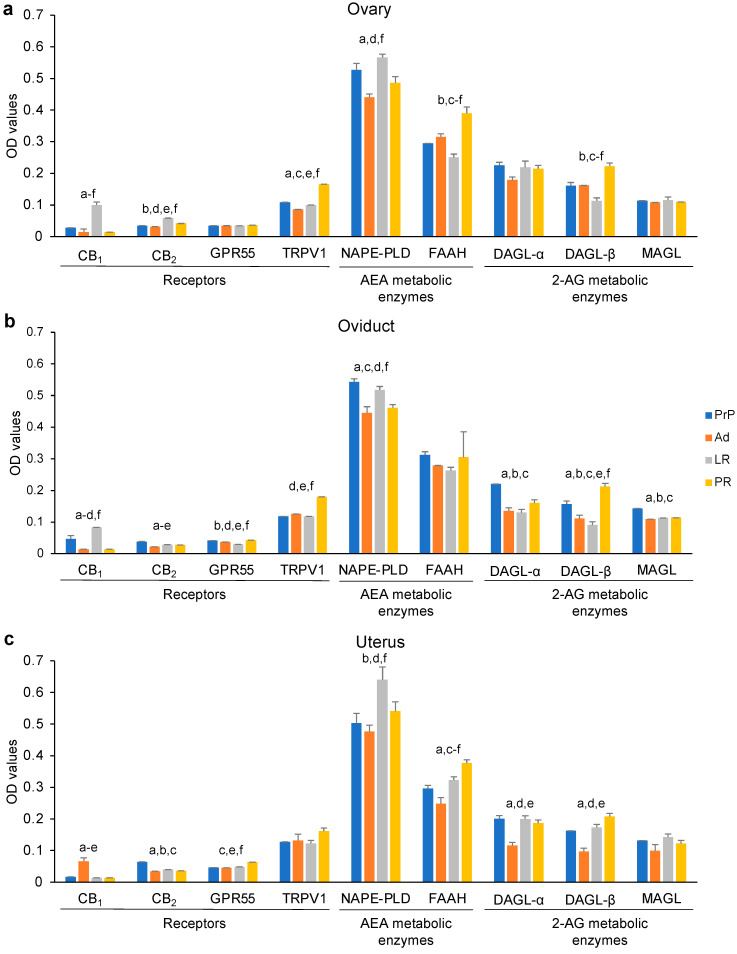
Expression of ECS components in female mouse reproductive tract during aging. Protein expression levels of ECS receptors (CB_1_, CB_2_, GPR55, TRPV1), AEA metabolic enzymes (NAPE-PLD, FAAH), and 2-AG metabolic enzymes (DAGL-α, DAGL-β, MAGL) detected in ovaries (**a**), oviducts (**b**), and uteri (**c**) collected from prepubertal (PrP), adult (Ad), late reproductive (LR), and post-reproductive (PR) mice. Results of three independent experiments expressed as mean OD values ± standard error of the mean (SEM). Letters indicate statistical significance (*p* < 0.05) as follows: a: PrP vs. Ad, b: PrP vs. LR; c: PrP vs. PR; d: Ad vs. LR; e: Ad vs. PR; f: LR vs. PR. For more detailed information, please refer to Appendix A.

**Figure 2 ijms-24-07542-f002:**
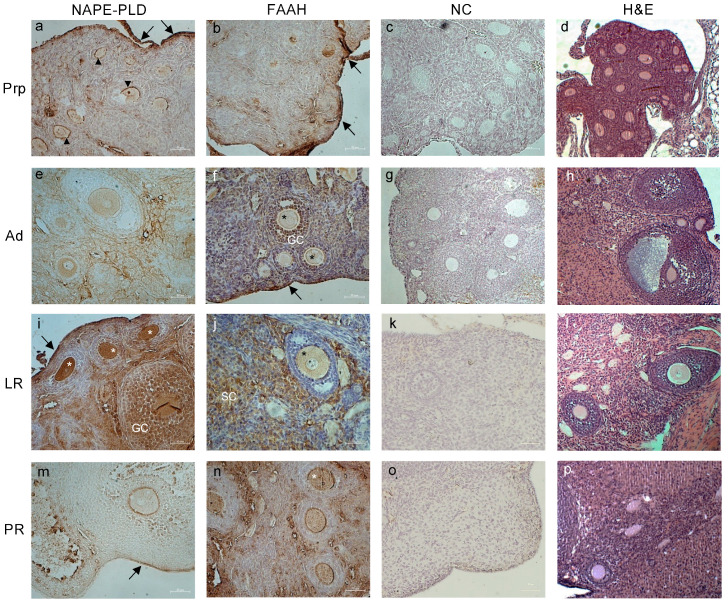
Representative images of tissue localization of NAPE-PLD and FAAH in mice ovaries during reproductive aging. Representative images of NAPE-PLD (**a**,**e**,**i**,**m**) and FAAH (**b**,**f**,**j**,**n**) immunoreactivity in prepubertal (PrP), adult (Ad), late reproductive (LR), and post-reproductive (PR) mouse ovaries. Negative controls (NC; (**c**,**g**,**k**,**o**)) and H&E staining (**d**,**h**,**l**,**p**) are also presented. Annotations and abbreviations—Arrows: ovarian surface epithelium; arrowheads: interstice granulosa cells–oocyte; asterisks: oocytes; GC: granulosa cells; SC: stroma cells. Scale bar: 50 µm. Magnification: ×200.

**Figure 3 ijms-24-07542-f003:**
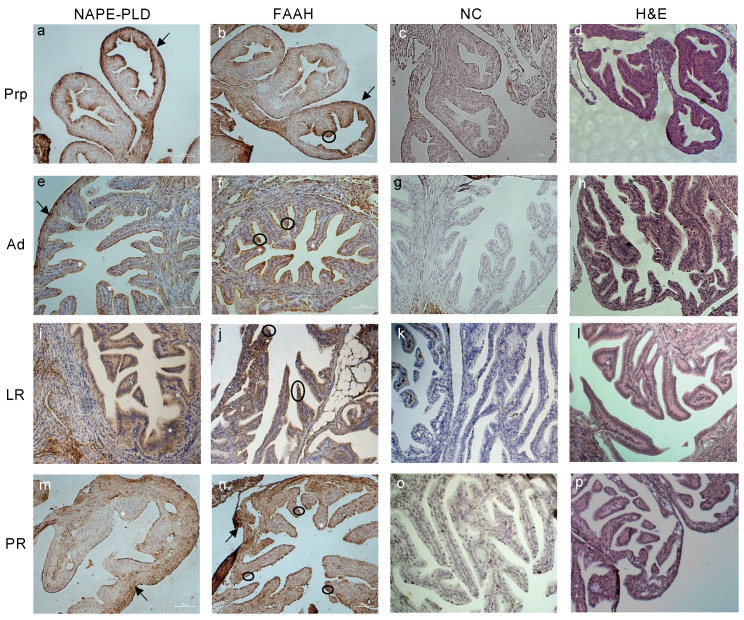
Representative images of tissue localization of NAPE-PLD and FAAH in mice oviducts during reproductive aging. Representative images of NAPE-PLD (**a**,**e**,**i**,**m**) and FAAH (**b**,**f**,**j**,**n**) immunoreactivity in prepubertal (PrP), adult (Ad), late reproductive (LR), and post-reproductive (PR) mouse oviducts. Negative controls (NC; (**c**,**g**,**k**,**o**)) and H&E staining (**d**,**h**,**l**,**p**) are also presented. Annotations—Arrows: serosa; circle: secretory cells; asterisks: ciliated cells. Scale bar: 50 µm. Magnification: ×200.

**Figure 4 ijms-24-07542-f004:**
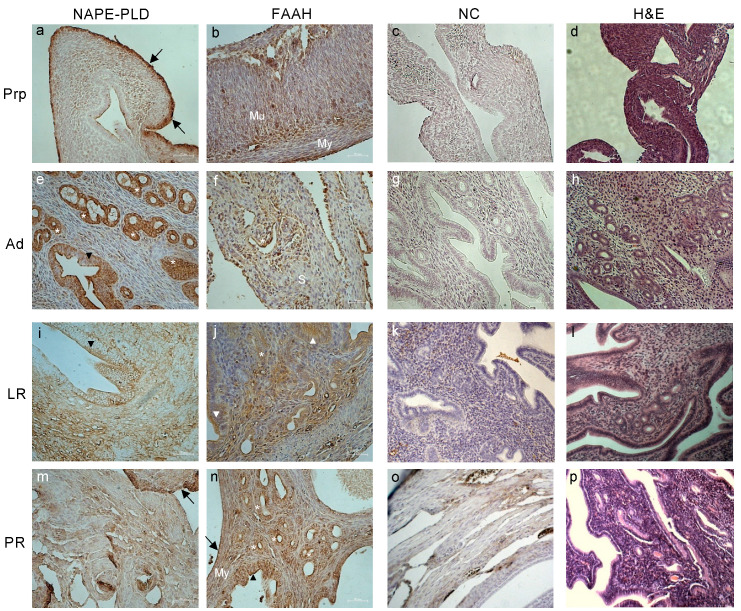
Representative images of tissue localization of NAPE-PLD and FAAH in mice uteri during reproductive aging. Representative images of NAPE-PLD (**a**,**e**,**i**,**m**) and FAAH (**b**,**f**,**j**,**n**) immunoreactivity in prepubertal (PrP), adult (Ad), late reproductive (LR), and post-reproductive (PR) mouse uteri. Negative controls (NC; (**c**,**g**,**k**,**o**)) and H&E staining (**d**,**h**,**l**,**p**) are also presented. Annotations and abbreviations—Arrows: perimetrium; arrowheads: columnar epithelium; asterisks: glands; My: myometrium; Mu: mucinous compartment; S: stroma. Scale bar: 50 µm. Magnification: ×200.

**Figure 5 ijms-24-07542-f005:**
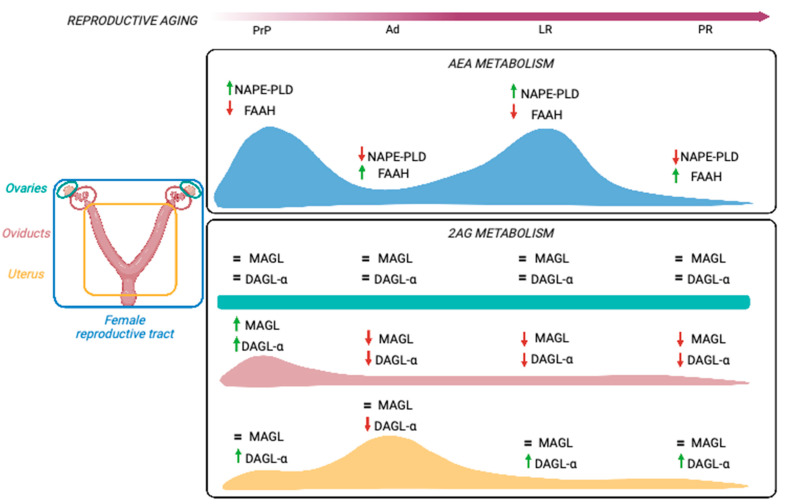
Schematic representation of age-related modulation of ECS in the female reproductive tract. A similar regulation of AEA metabolic enzymes is apparent in all organs (blue curve), whereas 2-AG metabolism seems to be differentially affected in an organ-specific manner (green: ovary, pink: oviduct, yellow: uterus). Abbreviations and annotation: Ad: adult; DAGL-α: diacylglycerol lipase α; FAAH: fatty acid amide hydrolase; LR: late reproductive; MAGL: monoacylglycerol lipase; NAPE-PLD: *N*-acylphosphatidylethanolamine-specific phospholipase D; PR: post-reproductive; PrP: prepubertal; pink arrow: reproductive aging timeline; green arrow: upregulation; red arrow: downregulation; =: unchanged.

## Data Availability

The data presented in this study are available in the present article and available on request to the corresponding author.

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
