# Peer review of "Endocannabinoid System Components of the Female Mouse Reproductive Tract Are Modulated during Reproductive Aging"

_ijms, 2023, doi:10.3390/ijms24087542_

Round 1

Reviewer 1 Report

This work is devoted to describing the dynamics of the expression of the components of the endocannabinoid signaling system in the organs of the reproductive system of mice of different ages. The study is focused on the detection of protein molecules of receptors and enzymes by ELISA and immunohistochemistry, therefore it is of high value compared to transcript analysis. The authors not only confirmed the fact of expression of the components of the endocannabinoid system in the ovary, oviducts, and uterus, but also revealed quantitative changes in the expression of both proteins and enzymes as reproductive tissues age.

The work is well illustrated and performed at a fairly high methodological level. The scientific reliability of the presented data is not in doubt. The conclusions are formulated quite neatly and, from my point of view, are justified.

In my opinion, the article can be published as it is.

Author Response

This work is devoted to describing the dynamics of the expression of the components of the endocannabinoid signaling system in the organs of the reproductive system of mice of different ages. The study is focused on the detection of protein molecules of receptors and enzymes by ELISA and

immunohistochemistry, therefore it is of high value compared to transcript analysis. The authors not only confirmed the fact of expression of the components of the endocannabinoid system in the ovary, oviducts, and uterus, but also revealed quantitative changes in the expression of both proteins and enzymes as reproductive tissues age. The work is well illustrated and performed at a fairly high methodological level. The scientific reliability of the presented data is not in doubt. The conclusions are formulated quite neatly and, from my point of view, are justified.

In my opinion, the article can be published as it is.

R: We thank the Referee for her/his kind appreciation of our work.

Reviewer 2 Report

The manuscript by Rossi et al. is a well-designed work to evaluate the correlation between endocannabinoid system components and aging in an animal model. The results are interesting and well-presented. I have just some minor questions:

- what is the possible meaning of a high level of NAPE-PLD in PrP ovary and oviducts?

- is there a cross-talk between the endocannabinoid system and the other paracrine and endocrine factors? Maybe the authors can mention something about that in the discussion.

Author Response

The manuscript by Rossi et al. is a well-designed work to evaluate the correlation between endocannabinoid system components and aging in an animal model. The results are interesting and well-presented. I have just some minor questions:

- what is the possible meaning of a high level of NAPE-PLD in PrP ovary and oviducts?

R1: We can speculate that high NAPE-PLD at a young age warrants a high concentration of its product, anandamide. At later stages modulation of the anandamide-hydrolase FAAH will reduce such a concentration to generate a suitable anandamide gradient that allows travel of the embryo through the oviduct and towards the uterus, as elegantly demonstrated by Dey’s group in 2006 (Wang H, Xie H, Guo Y, Zhang H, Takahashi T, Kingsley PJ, Marnett LJ, Das SK, Cravatt BF, Dey SK. Fatty acid amide hydrolase deficiency limits early pregnancy events. J Clin Invest. 2006 Aug;116(8):2122-31).

This hypothesis has been now included in the amended Discussion, along with the pertinent reference (see lines 254-262, and reference 26). We thank the Referee for raising this interesting point.

- is there a cross-talk between the endocannabinoid system and the other paracrine and endocrine factors? Maybe the authors can mention something about that in the discussion.

R2: Several cross-talks have been indeed reported between the endocannabinoid system and other paracrine and endocrine factors that regulate female reproduction, as recently reviewed by our group (Cecconi S, Rapino C, Di Nisio V, Rossi G, Maccarrone M. The (endo)cannabinoid signaling in female reproduction: What are the latest advances? Prog Lipid Res. 2020 Jan;77:101019). We have clarified this issue by enlarging the revised Discussion (see also reply to point 4 raised by Referee #3). The pertinent reference has been added to the amended reference list (see lines 277-282, and reference 34).

Reviewer 3 Report

I read with great interest the paper and I must admit that I've greatly appreciated the work of Rossi and co-authors.

I've underlined many strenghts in this work:

- the conceptualization is good, originality very good.

- The importance of this research for the field is high.

- Results are sounding.

Let's move on weaknesses:

- The description of results and subsequent discussion is hard to understand in many points. I know that authors obtained many data with many variables, but -please consider my comment as a positive suggestion- they can do better actually. I may suggest to improve the description considering the analysis of up- and downregulation of biosinthetic and degradative enzymes in different ages together and then move on receptors expression. This described in different organs. I believe it will became simpler; summarizing: in this way we can easily discuss that AEA metabolism is clearly modulated in all tracts to have the minimum in reproductive age; 2-AG synthesis is more organo-specific, even if it could be noted that FAAH also participates in 2-AG catabolism and it explains in part why MAGL modulation is not mirrored by DAGL acrivation. CB1r expression in uterus is interesting and may be explained taking into consideration hormone cross-talking not yet elucidated. Authors tried to explain the situation in discussion and I invite them to enlarge this interesting part.

Minor points: plaese provide more details on used antibodies and describe if a normalization of protein quantification in customized ELISA essays was made. Did authors perform a calibration curve with available standard?

Author Response

I read with great interest the paper and I must admit that I've greatly appreciated the work of Rossi and co-authors. I've underlined many strenghts in this work:

- the conceptualization is good, originality very good.

- The importance of this research for the field is high.

- Results are sounding.

R1: We thank the Referee for her/his kind appreciation of our work.

Let's move on weaknesses:

- The description of results and subsequent discussion is hard to understand in many points. I know that authors obtained many data with many variables, but -please consider my comment as a positive suggestion- they can do better actually. I may suggest to improve the description considering the analysis of up- and downregulation of biosinthetic and degradative enzymes in different ages together and then move on receptors expression. This described in different organs. I believe it will became simpler; summarizing: in this way we can easily discuss that AEA metabolism is clearly modulated in all tracts to have the minimum in reproductive age; 2-AG synthesis is more organospecific, even if it could be noted that FAAH also participates in 2-AG catabolism and it explains in part why MAGL modulation is not mirrored by DAGL acrivation.

R2: We thank the Referee for the useful suggestion. However, the suggested reorganization would have modified the whole flow of thoughts on which the discussion has been based. Nevertheless, we understand and agree on the need of a summarizing recap of the findings and discussion points on the potential mechanisms of ECS regulation in female reproductive tract during aging. To this end, a new summarizing paragraph and a new figure 5 have been included in the revised Discussion (lines 305-311, and figure 5).

CB1r expression in uterus is interesting and may be explained taking into consideration hormone cross-talking not yet elucidated. Authors tried to explain the situation in discussion and I invite them to enlarge this interesting part.

R3: Several cross-talks have been indeed reported between the endocannabinoid system and other hormones that regulate female reproduction, as recently reviewed by our group (Cecconi S, Rapino C, Di Nisio V, Rossi G, Maccarrone M. The (endo)cannabinoid signaling in female reproduction: What are the latest advances? Prog Lipid Res. 2020 Jan;77:101019). We have clarified this issue by enlarging the revised Discussion (see also reply to point 2 raised by Referee #2). The pertinent reference has been added to the amended reference list (see lines 277-282, and reference 34).

Minor points:

- plaese provide more details on used antibodies and describe if a normalization of protein quantification in customized ELISA essays was made. Did authors perform a calibration curve with available standard?

R4: We have clearly indicated the catalogue numbers of the used antibodies in the revised version (lines 322, 324, 328-334), and apologize for not having done it before. As for the normalization/calibration of the ELISA assays, we did perform dose-response experiments by using different amounts (0 – 2.5 – 5.0 - 10 μg/well) of (Ad) mouse uterus extracts with each antibody. A linear dependence on the amount of sample was found with all antibodies. We have better clarified this procedure in the revised Materials and Methods section (lines 368-371). We would like to remind that unfortunately none of the proteins that we have measured (CB1, CB2, GPR55, TRPV1, NAPE-PLD, FAAH, DAGL-α, DAGL-β, MAGL) are available as purified standards.

Reviewer 4 Report

In my opinon this is an interesting manuscript focused on evaluating the expression and implication of ECS (CB receptors, endocannabinoids and the enzymes respnsible for tehir synthesis and degradation) in reproductive orgarns.  I consider that this manuscript it is sutable for publication in IJMS. The manuscript is clear and well written.  Just a few aspects to be revised/considered by the authors: 

- I suggest explaining the abbreviations of PrP, LR, Ad, and PR the fisrt time that appear in the mannuscript. I think it is in Figure 1. They are explained in methods but in this case results are shown before. This change would be useful for the readers. 

- Just more than a minor comment: In Figure 1 the authors indicate significance at least p<0.05. I suggest deleting the word "at least". In my opinon sounds better. This is just a suggestion. 

- In Figure 1A and B I think it would be useful to have the X axis legend also. 

-  In Figure 2-4 in some images it is difficult to see the symbols (arrows, cicles..) Did the author check to include them black colored instead of white? Maybe in this case they would be more visible. 

- I missed more information about the preexisting data ( ECS and reproduction) I suggest the authors including this. 

Author Response

In my opinon this is an interesting manuscript focused on evaluating the expression and implication of ECS (CB receptors, endocannabinoids and the enzymes respnsible for tehir synthesis and degradation) in reproductive orgarns. I consider that this manuscript it is sutable for publication in IJMS.

R1: We thank the Referee for her/his kind appreciation of our work.

The manuscript is clear and well written. Just a few aspects to be revised/considered by the authors:

- I suggest explaining the abbreviations of PrP, LR, Ad, and PR the fisrt time that appear in the mannuscript. I think it is in Figure 1. They are explained in methods but in this case results are shown before. This change would be useful for the readers.

R2: Done as suggested. See lines 94-95 of the amended manuscript.

- Just more than a minor comment: In Figure 1 the authors indicate significance at least p<0.05. I suggest deleting the word "at least". In my opinon sounds better. This is just a suggestion.

R3: Done as suggested. See line 105 of the amended manuscript.

- In Figure 1A and B I think it would be useful to have the X axis legend also.

R4: Done as suggested. See Figure 1 included in the amended manuscript.

- In Figure 2-4 in some images it is difficult to see the symbols (arrows, cicles..) Did the author check to include them black colored instead of white? Maybe in this case they would be more visible.

R5: We modified the figures as suggested, whenever the black colour would help the visualization of the figure annotation. See Figures 2-4 included in the amended manuscript.

- I missed more information about the preexisting data (ECS and reproduction) I suggest the authors including this.

R6: More information has been included in the Introduction, as suggested. See lines 49-54, 62-64, 71-74, 76-79 and new references 8, 9, 14-17 of the amended manuscript.